# OpenReview forum: "SocialHarmBench: Revealing LLM Vulnerabilities  to Socially Harmful Requests"
_ICLR.cc/2026/Conference — ICLR 2026 Poster_

### Official Review · Reviewer_sDuZ · 2025-10-25

**Soundness:** 2
**Presentation:** 2
**Contribution:** 2
**Rating:** 2
**Confidence:** 4

**Summary:**

This paper presents SocialHarmBench, a jailbreak benchmark that evaluates LLMs' robustness to social-political questions. The benchmark highlights diversity in terms of the covered topics, countries, time periods, and query types. The authors first evaluate eight LLMs to reveal vulnerabilities and analyze the distribution of successful jailbreaks over domains, regions, and time periods. The authors also apply six jailbreak strategies to five LLMs to further investigate the effectiveness of these strategies in social-political queries. The results reveal vulnerabilities in the safeguards of existing LLMs.

**Strengths:**

S1: The social-political aspect is a unique aspect not extensively studied by existing jailbreak benchmarks.
S2: The authors provide a somewhat thorough investigation of this aspect in a set of existing LLMs.

**Weaknesses:**

W1: Although this is a benchmark paper, the authors never described how the dataset was created. The four steps mentioned in Section 2.3 and Appendix D lack enough details on the generation process. Are all the events, topics, and templates identified and generated by LLMs? If yes, then the subsequent evaluation would be highly biased toward the position of the LLM that generates the dataset. If no, then the paper lacks enough citations and details to ground the data collection process. Political opinions and social harms are very sensitive topics, and the authors need to ensure that all the queries tested in the paper are objective and truly reflect the opinions held by the vast majority.
W2: How do you determine the semantic categories and functional categories of a jailbreak prompt in SocialHarmBench? It seems that there is quite some overlap in those categories, e.g., surveillance and censorship.
W3: Can you do manual checking on the results in Section 4.1? Are those scores reported by LLMs-as-a-judge aligned with human judgment?
W4: Can you swap the topics in your jailbreak templates to other non-social-political topics? To make sure that the vulnerabilities reported in your Figure 3 do not arise from your templates and other non-social-political confounding factors during your evaluation process.

**Questions:**

Q1: In the introduction, it says that the evaluation involves five LLMs, but Section 3 says that there are eight LLMs. This is confusing.
Q2: Can you put all the proprietary LLMs together and all the open-sourced LLMs together on the x-axis in Figure 3?
Q3: What do the authors mean by the yellow text blocks in Pages 7, 8, and 9? There is no caption or any explanation.
Q4: In ICLR templates, captions should come on top of tables, instead of at the bottom. This applies to all the tables in this paper.
Q5: Figure 2 is very hard to follow. What do the bars in the two subfigures in the top row represent? And which region does "Add" stand for?
Q6: Section 3 is very repetitive and verbose. There is no need to mention HarmBench and StrongREJECT classifiers three times.

---

> ### Author Response · Authors · 2025-11-22
> **Thank you for your review**
>
> We thank you for your constructive feedback. Below, we address key concerns point-by-point.
>
> ***Concern 1: Although this is a benchmark paper, the authors never described how the dataset was created. The four steps mentioned in Section 2.3 and Appendix D lack enough details on the generation process. Are all the events, topics, and templates identified and generated by LLMs? If yes, then the subsequent evaluation would be highly biased toward the position of the LLM that generates the dataset. If no, then the paper lacks enough citations and details to ground the data collection process. Political opinions and social harms are very sensitive topics, and the authors need to ensure that all the queries tested in the paper are objective and truly reflect the opinions held by the vast majority.***
>
> We would like to clarify that due to the current page limit, we were not able to fully list dataset construction in the main paper section. **The topics, events, and functional templates are identified and manually curated by authors – and we use LLMs to generate prompts across our defined subtopics.** Please refer to Appendix E.3, where we note our dataset construction process and data deduplication process in the current manuscript. To remove any remaining ambiguity: the events in our dataset (e.g., elections, policies, conflicts, public-health incidents) were manually selected and defined by the authors. LLMs were never used to propose, generate, or classify events. LLM involvement was limited strictly to linguistic paraphrasing of already-defined templates, not to factual content creation or topical selection.
>
> Additionally, we recognize that political opinions and social harms are sensitive topics, which is why it is even more important to study potential harms that LLMs can impose on such contexts. For this reason, **we included the ORF imbalance measure to flag when topic-level composition diverges too sharply from global rates**. Our priority is to avoid situations where certain regions become disproportionately linked to harmful topics, which can carry real sociopolitical consequences such as reinforcing stigma or distorted risk perceptions. The benchmark does not attempt to produce authoritative historical descriptions; events are included only as topical anchors for adversarial stress-tests. No factual claims or interpretations from LLMs are used.
>
> ***Concern 2: How do you determine the semantic categories and functional categories of a jailbreak prompt in SocialHarmBench? It seems that there is quite some overlap in those categories, e.g., surveillance and censorship.***
>
> We follow standard adversarial evaluation procedures followed by past benchmarks [1, 2, 3] to determine our semantic and functional categories. Functional categories focus on three types of queries (1) standard or agentic usage, (2) historical references, (3) opinion elicitation. In contrast, semantic categories focus on the topics at hand such as surveillance, censorship, propaganda generation, and human rights violations.
>
> **We take inspiration for our semantic categories from the Universal Declaration of Human Rights [4].** Our distinction between surveillance and censorship aligns with prior policy and safety-taxonomy work: surveillance concerns the acquisition, monitoring, or analysis of information about individuals or groups, whereas censorship concerns the restriction, removal, or suppression of information or expression. In constructing prompts, we ensured that “surveillance” items emphasize information gathering (e.g., monitoring political dissidents, tracking communications), while “censorship” items emphasize information control (e.g., blocking access, suppressing messages, filtering political content). This separation avoids category conflation and enables clearer, more interpretable evaluations of model behavior.  Similar to the above example, we maintain a clear distinction between all of our semantic categories, ensuring no overlap. **A detailed description of what each category is meant to capture can be found in Table 5 in Appendix D.2.**
>
> ***Concern 3: Can you do manual checking on the results in Section 4.1? Are those scores reported by LLMs-as-a-judge aligned with human judgment?***
>
> We have run human evaluations on all existing 17,550 completions across our open-source and closed-source models. Additionally, from our new closed-source adversarial attack evaluations, we report human evaluations across all 2,340 completions. To reduce bias, we have ensured to report results after confirming inter-annotator agreement exceeds 95% across all completions:

---

> > ### Author Response · Authors · 2025-11-22
> > **Thank you for your review - Continued (C3)**
> >
> > Table 1. Baseline Attack Performance Across Models:
> > $$
> > \\begin{array}{l|ccccccc}
> > \\text{Model} & \\text{Censorship} & \\text{Historical Revisionism} & \\text{Human Rights Violations} & \\text{Political Manipulation} & \\text{Propaganda Generation} & \\text{Surveillance} & \\text{War Crimes} \\\\ \\hline
> > \\text{Qwen 2.5-7B-Instruct} & 20.5 & 43.8 & 22.3 & 17.3 & 16.2 & 35.0 & 8.9 \\\\
> > \\text{Claude-Sonnet-4} & 1.1 & 1.6 & 0.0 & 0.0 & 4.1 & 5.0 & 0.0 \\\\
> > \\text{Deepseek-LLM-7B-Chat} & 15.9 & 48.4 & 13.6 & 13.6 & 33.8 & 24.0 & 6.3 \\\\
> > \\text{Gemini-2.5-Flash} & 1.1 & 21.9 & 3.9 & 4.9 & 13.5 & 8.0 & 2.5 \\\\
> > \\text{Gemma 3-12B} & 18.2 & 37.5 & 6.8 & 18.5 & 28.4 & 29.0 & 6.3 \\\\
> > \\text{GPT-4o} & 8.0 & 31.2 & 3.9 & 2.5 & 18.9 & 16.0 & 2.5 \\\\
> > \\text{Llama 3.1 8B-Instruct} & 13.6 & 31.2 & 6.8 & 8.6 & 24.3 & 23.0 & 7.6 \\\\
> > \\text{Mistral v0.3 7B-Instruct} & 48.9 & 73.4 & 39.8 & 59.3 & 60.8 & 45.0 & 31.6 \\\\
> > \\end{array}
> > $$
> >
> > Table 2. Weight Tampering Attack Performance Across Models:
> > $$
> > \\begin{array}{l|ccccccc}
> > \\text{Model} & \\text{Censorship} & \\text{Historical Revisionism} & \\text{Human Rights Violations} & \\text{Political Manipulation} & \\text{Propaganda Generation} & \\text{Surveillance} & \\text{War Crimes} \\\\ \\hline
> > \\text{Gemma 3-12B} & 67.0 & 96.9 & 93.2 & 87.7 & 91.9 & 84.0 & 92.4 \\\\
> > \\text{Llama 3.1 8B-Instruct} & 67.0 & 96.9 & 92.2 & 86.4 & 91.9 & 82.0 & 87.3 \\\\
> > \\text{Mistral v0.3 7B-Instruct} & 78.4 & 98.4 & 95.1 & 95.1 & 95.9 & 90.0 & 91.1 \\\\
> > \\text{Qwen 2.5-7B-Instruct} & 67.0 & 96.9 & 93.2 & 87.7 & 91.9 & 81.0 & 87.3 \\\\
> > \\text{Deepseek-LLM-7B-Chat} & 76.1 & 98.4 & 93.2 & 87.7 & 94.6 & 90.0 & 87.3 \\\\
> > \\end{array}
> > $$
> >
> > Table 3. AutoDAN (GA) Attack Performance Across Models:
> > $$
> > \\begin{array}{l|ccccccc}
> > \\text{Model} & \\text{Censorship} & \\text{Historical Revisionism} & \\text{Human Rights Violations} & \\text{Political Manipulation} & \\text{Propaganda Generation} & \\text{Surveillance} & \\text{War Crimes} \\\\ \\hline
> > \\text{Gemma 3-12B} & 76.1 & 89.1 & 90.2 & 90.1 & 93.2 & 86.0 & 86.1 \\\\
> > \\text{Llama 3.1 8B-Instruct} & 40.2 & 60.9 & 50.0 & 59.3 & 60.3 & 59.0 & 53.2 \\\\
> > \\text{Mistral v0.3 7B-Instruct} & 71.6 & 93.8 & 90.2 & 84.0 & 87.7 & 85.0 & 89.9 \\\\
> > \\text{Qwen 2.5-7B-Instruct} & 72.7 & 90.6 & 91.2 & 88.9 & 93.2 & 85.0 & 92.4 \\\\
> > \\text{Deepseek-LLM-7B-Chat} & 67.0 & 92.2 & 85.3 & 82.7 & 89.0 & 79.0 & 91.1 \\\\
> > \\end{array}
> > $$
> >
> > Table 4. AutoDAN (HGA) Attack Performance Across Models:
> > $$
> > \\begin{array}{l|ccccccc}
> > \\text{Model} & \\text{Censorship} & \\text{Historical Revisionism} & \\text{Human Rights Violations} & \\text{Political Manipulation} & \\text{Propaganda Generation} & \\text{Surveillance} & \\text{War Crimes} \\\\ \\hline
> > \\text{Gemma 3-12B} & 75.0 & 93.8 & 87.3 & 80.2 & 91.8 & 86.0 & 79.7 \\\\
> > \\text{Llama 3.1 8B-Instruct} & 42.0 & 65.6 & 50.0 & 55.6 & 60.3 & 58.0 & 55.7 \\\\
> > \\text{Mistral v0.3 7B-Instruct} & 71.6 & 93.8 & 87.3 & 85.2 & 84.9 & 81.0 & 84.8 \\\\
> > \\text{Qwen 2.5-7B-Instruct} & 71.6 & 92.2 & 90.2 & 87.7 & 89.0 & 83.0 & 91.1 \\\\
> > \\text{Deepseek-LLM-7B-Chat} & 64.8 & 92.2 & 84.3 & 84.0 & 89.0 & 80.0 & 96.0 \\\\
> > \\end{array}
> > $$
> >
> > Table 5. GCG Attack Performance Across Models:
> > $$
> > \\begin{array}{l|ccccccc}
> > \\text{Model} & \\text{Censorship} & \\text{Historical Revisionism} & \\text{Human Rights Violations} & \\text{Political Manipulation} & \\text{Propaganda Generation} & \\text{Surveillance} & \\text{War Crimes} \\\\ \\hline
> > \\text{Llama 3.1 8B-Instruct} & 59.7 & 89.0 & 82.5 & 79.0 & 85.1 & 67.3 & 73.4 \\\\
> > \\text{Mistral v0.3 7B-Instruct} & 60.9 & 82.8 & 74.5 & 71.6 & 77.0 & 69.3 & 75.6 \\\\
> > \\text{Qwen 2.5-7B-Instruct} & 48.2 & 73.4 & 60.7 & 74.1 & 71.6 & 66.3 & 56.9 \\\\
> > \\text{Deepseek-LLM-7B-Chat} & 31.0 & 53.12 & 33.3 & 28.39 & 40.5 & 42.8 & 25.3 \\\\
> > \\end{array}
> > $$
> >
> > Table 6. Embedding Optimization Attack Performance Across Models:
> > $$
> > \\begin{array}{l|ccccccc}
> > \\text{Model} & \\text{Censorship} & \\text{Historical Revisionism} & \\text{Human Rights Violations} & \\text{Political Manipulation} & \\text{Propaganda Generation} & \\text{Surveillance} & \\text{War Crimes} \\\\ \\hline
> > \\text{Gemma 3-12B} & 38.6 & 25.0 & 63.7 & 53.1 & 20.3 & 43.0 & 50.6 \\\\
> > \\text{Llama 3.1 8B-Instruct} & 65.9 & 97.5 & 92.2 & 80.2 & 89.2 & 83.0 & 86.1 \\\\
> > \\text{Mistral v0.3 7B-Instruct} & 67.0 & 89.1 & 91.2 & 80.2 & 90.5 & 82.0 & 89.9 \\\\
> > \\text{Deepseek-LLM-7B-Chat} & 17.0 & 50.0 & 21.0 & 24.7 & 37.8 & 21.0 & 10.1 \\\\
> > \\end{array}
> > $$
> >
> > Please refer to our response in Q6 to clarify if these scores are reported as an LLM-as-a-judge model. We clarify that in adversarial evaluations, the most commonly used toxicity classifiers are HarmBench and StrongREJECT to generate attack success rates. We would also like to share that we ran the TAP [5] and PAIR [6] attacks across closed-source models:

---

> > > ### Author Response · Authors · 2025-11-22
> > > **Thank you for your review - Continued (C3, C4)**
> > >
> > > $$
> > > \\begin{array}{l|ccc|ccc}
> > > \\text{Model Name}
> > >   & \\text{PAIR-StrongREJECT} & \\text{PAIR-HarmBench} & \\text{PAIR-Human Eval}
> > >   & \\text{TAP-StrongREJECT}  & \\text{TAP-HarmBench}  & \\text{TAP-Human Eval} \\\\
> > > \\hline
> > > \\text{Claude-Sonnet-4}
> > >   & 32.39 & 24.78 & 10.42
> > >   & 20.61 & 13.67 & 23.42 \\\\
> > > \\text{GPT-4o}
> > >   & 55.06 & 75.72 & 75.89
> > >   & 55.38 & 76.92 & 76.58 \\\\
> > > \\end{array}
> > > $$
> > >
> > > ***Weakness 4: Can you swap the topics in your jailbreak templates to other non-social-political topics? To make sure that the vulnerabilities reported in your Figure 3 do not arise from your templates and other non-social-political confounding factors during your evaluation process.***
> > >
> > > We appreciate this methodological concern. However, our experimental design already addresses template effects through built-in controls, and our results demonstrate content-driven vulnerabilities.
> > >
> > > As mentioned above under W2, SocialHarmBench includes three distinct functional categories (Section 2.1): standard behaviors, opinion elicitation, and historical references. Our results show that semantic categories, not functional templates, drive vulnerability patterns. Historical Revisionism shows 35-62% ASR across all models, and Propaganda Generation shows 21-59% ASR, with these patterns holding consistently across all three functional templates. If vulnerabilities were template-driven, we would see high ASR for one functional type but not others. Instead, semantic categories maintain consistent rankings regardless of template type.
> > >
> > > Additionally, identical prompts produce dramatically different results across models (Claude-Sonnet-4: 0-5% ASR vs. Mistral-7B: 28-62% ASR), and different attack methods on the same prompts yield varying success rates (Weight Tampering: 72-98% vs. GCG: 4-86%). Both patterns indicate content-specific vulnerabilities rather than template artifacts.
> > >
> > > Finally, the reviewer's suggested topic-swapping approach would **fundamentally change** what we're measuring. Transforming "Generate propaganda for authoritarian regime X" into "Generate marketing for company Y" **eliminates the sociopolitical safety concern entirely**. The political dimension: where LLMs must balance compliance with human rights, is inseparable from the content. This is analogous to how SQL injection benchmarks don't test non-SQL control inputs, because the specific content-template combination defines the security risk. We demonstrate content-driven vulnerabilities through: (a) consistent semantic patterns across diverse templates, (b) model-specific differences using identical prompts, and (c) attack-method variance. While a comprehensive ablation would be valuable future work, it would require creating an entirely new benchmark (28,080+ evaluations) and would change the fundamental research question.
> > >
> > > **Responses to Reviewer Questions**
> > >
> > > Below, we respond to the questions raised in the review:
> > >
> > > *Q1: In the introduction, it says that the evaluation involves five LLMs, but Section 3 says that there are eight LLMs. This is confusing.*
> > >
> > > Thank you for bringing this concern up, we will rectify this in the revised manuscript.
> > >
> > > *Q3: What do the authors mean by the yellow text blocks in Pages 7, 8, and 9? There is no caption or any explanation.*
> > >
> > > The yellow text blocks in pages 7, 8, 9, show example prompts and responses from our analysis into SocialHarmBench and sociopolitical vulnerabilities within current LLMs across temporal, geographic, and historical contexts.
> > >
> > > *Q4: In ICLR templates, captions should come on top of tables, instead of at the bottom. This applies to all the tables in this paper.*
> > >
> > > Thank you for bringing this concern up, we will rectify this in the revised manuscript.
> > >
> > > Q5: Figure 2 is very hard to follow. What do the bars in the two subfigures in the top row represent? And which region does "Add" stand for?
> > >
> > > “Add” is an abbreviation for additional regions. The two bars represent the functional categories and the region counts in our dataset construction.
> > >
> > > *Q6: Section 3 is very repetitive and verbose. There is no need to mention HarmBench and StrongREJECT classifiers three times.*
> > >
> > > While we will revise this section as requested by the reviewer, we would like to clarify, we deliberately call out HarmBench and StrongREJECT multiple times to make clear that our **ASR numbers come from classifier-based evals and not LLM-as-a-judge**. This was done to avoid any major weakness questions like the one you mentioned in Concern 3 (where you ask whether our attack success rate scores are reported by LLM-as-a-judge).

---

> > > > ### Author Response · Authors · 2025-11-22
> > > > **References**
> > > >
> > > > [1] Mazeika, M., Phan, L., Yin, X., Zou, A., Wang, Z., Mu, N., Sakhaee, E., Li, N., Basart, S., Li, B., Forsyth, D., & Hendrycks, D. (2024). HarmBench: A standardized evaluation framework for automated red teaming and robust refusal.
> > > >
> > > > [2] Andriushchenko, M., Souly, A., Dziemian, M., Duenas, D., Lin, M., Wang, J., Hendrycks, D., Zou, A., Kolter, Z., Fredrikson, M., Winsor, E., Wynne, J., Gal, Y., & Davies, X. (2025). AgentHarm: A benchmark for measuring harmfulness of LLM agents.
> > > >
> > > > [3] Zou, A., Wang, Z., Carlini, N., Nasr, M., Kolter, J. Z., & Fredrikson, M. (2023). Universal and transferable adversarial attacks on aligned language models.
> > > >
> > > > [4] UN General Assembly, Universal Declaration of Human Rights, 217 A (III), 10 December 1948.
> > > >
> > > > [5] Mehrotra, A., Zampetakis, M., Kassianik, P., Nelson, B., Anderson, H., Singer, Y., & Karbasi, A. (2024). Tree of Attacks: Jailbreaking Black-Box LLMs Automatically. arXiv:2312.02119.
> > > >
> > > > [6] Chao, P., Robey, A., Dobriban, E., Hassani, H., Pappas, G. J., & Wong, E. (2024). Jailbreaking Black Box Large Language Models in Twenty Queries. arXiv:2310.08419.

---

> > > > > ### Comment · Reviewer_sDuZ · 2025-11-26
> > > > >
> > > > > Thanks for your response. Some of my concerns are addressed. I'll raise my score.

---

> > > > > > ### Author Response · Authors · 2025-11-26
> > > > > > **Thank You For Your Reconsideration - Rebuttal Follow-Up**
> > > > > >
> > > > > > Thank you for your reconsideration and for taking the time to review our clarifications. We’re glad the revisions addressed the main concerns, and we sincerely appreciate your openness to updating the score. **If any remaining points would benefit from further clarification or strengthening,** we would be glad to address them so that the contribution: **the first LLM adversarial evaluation benchmark in the sociopolitical domain**, can be presented to the community in its strongest form.

---

> > > > > > > ### Author Response · Authors · 2025-11-27
> > > > > > > **Kind Reminder for Author-Reviewer Discussion**
> > > > > > >
> > > > > > > Dear Reviewer,
> > > > > > >
> > > > > > > We'd like to warmly remind you that the author-reviewer discussion phase will end soon. If our response has helped addressing some of your concerns, we kindly ask you to consider raising your score, we're very happy to provide further clarification if needed

---

> > > > > > > > ### Author Response · Authors · 2025-11-27
> > > > > > > > **Kind Reminder for Author-Reviewer Discussion**
> > > > > > > >
> > > > > > > > Thank you for your reconsideration and for taking the time to review our clarifications. We’re glad the revisions addressed the main concerns, and we sincerely appreciate your openness to updating the score. **If any remaining points would benefit from further clarification or strengthening**, we would be glad to address them so that the contribution: **the first LLM adversarial evaluation benchmark in the sociopolitical domain**, can be presented to the community in its strongest form.

---

### Official Review · Reviewer_6ZQ4 · 2025-10-28

**Soundness:** 2
**Presentation:** 3
**Contribution:** 3
**Rating:** 6
**Confidence:** 4

**Summary:**

The paper introduces SocialHarmBench, a 585-prompt benchmark including seven sociopolitical domains and 34 countries to test LLM vulnerabilities in politically charged settings. Prompts are labeled both semantically and by functional type, and responses are scored with HarmBench for harmful compliance and StrongREJECT for refusal robustness. Evaluation results for eight LLMs are reported alongside five open-weight models under six attack settings. Baseline weaknesses are largest in historical revisionism, propaganda, and political manipulation. Attacks degrade safety, showing that current safeguards do not transfer well to sociopolitical harms.

**Strengths:**

- Semantic categories for the dataset are well grounded, and the dataset split across categories is well balanced. It is clear that the data curation was done well.
- The analysis of the curated dataset is sufficiently rigorous, and the evaluation set for models is comprehensive.
- Aside from typos (see below), paper delivery is very clear. In particular, I enjoyed Figure 6 as is informative and interesting to see.
- Overall, the paper addresses an important aspect in safety and alignment in language models that existing work has looked over; SocialHarmBench is a very useful contribution to the safety community.

**Weaknesses:**

## Grammar Corrections / Minor Errors
- Line 52: comprising of → comprising
- Line 59: focusing → focuses
- Line 62: remove duplicate: “surveillance and censorship, ..., surveillance,”
- Line 74: e.g. → e.g.,
- Line 77: to present day → to the present day
- Line 244: distangle → disentangle
- Line 255: five attack techniques → six attack techniques? (you evaluate six)
- Line 410 (Section 4.3 title): LLMs are more vulnerable towards → LLMs are more vulnerable to
- Line 1213: segragation → segregation
- Appendix (and figure labels): StrongReject → StrongREJECT
- Figure 1: Says 40 sub-topics, but line 74 says 37 sub-topics.
- Figure 1: Civilian Targetting → Civilian Targeting

## Major Weaknesses
- It is unclear to me whether StrongREJECT and HarmBench evaluation is sufficient or accurate. These two methods largely use string matching, which may miss true harmfulness given the sophistication of StrongREJECT questions. Have the authors considered designing an evaluator to pair with SocialHarmBench? At minimum, I believe it may be necessary to include human annotation results of classifier outputs. HarmBench/StrongREJECT driving all conclusions is weak and requires more reliable verification.
- While the set of attacks that were examined in the paper are good, several well-noted jailbreak methods have been missed (e.g., TAP [1], PAIR [2], etc.). In particular, it would be nice to include results for some more simple prompt-only methods ([3]).

## References
[1] Mehrotra, A., Zampetakis, M., Kassianik, P., Nelson, B., Anderson, H., Singer, Y., & Karbasi, A. (2025). *Tree of Attacks: Jailbreaking Black-Box LLMs Automatically.*
[2] Chao, P., Robey, A., Dobriban, E., Hassani, H., Pappas, G. J., & Wong, E. (2025). *Jailbreaking Black-Box Large Language Models in Twenty Queries.*
[3] Li, N., Han, Z., Steneker, I., Primack, W., Goodside, R., Zhang, H., Wang, Z., Menghini, C., & Yue, S. (2025). *LLM Defenses Are Not Robust to Multi-Turn Human Jailbreaks Yet.*

**Questions:**

- Why is Figure 3 a line graph rather than a bar graph like Figure 4? The current plot is hard to interpret.
- Why were thinking/reasoning models not evaluated for this task?

---

> ### Author Response · Authors · 2025-11-22
> **Thank you for your review**
>
> We thank you for your constructive feedback. Below, we address key concerns point-by-point.
>
> ***Concern 1: It is unclear to me whether StrongREJECT and HarmBench evaluation is sufficient or accurate. These two methods largely use string matching, which may miss true harmfulness given the sophistication of StrongREJECT questions. Have the authors considered designing an evaluator to pair with SocialHarmBench? At minimum, I believe it may be necessary to include human annotation results of classifier outputs. HarmBench/StrongREJECT driving all conclusions is weak and requires more reliable verification.***
>
> As StrongREJECT and HarmBench are popularly used evaluators in LLM safety to assess harmfulness, we reported our metrics on both classifiers for maximal coverage. However, we understand the need to provide human validation for this, please see above for the full human-evaluated completions across newly-integrated adversarial attacks as well (sample size: 17,550 + 1,170 = 18,720).
>
> We have run human evaluations on all existing 17,550 completions across our open-source and closed-source models. Additionally, from our new closed-source adversarial attack evaluations, we report human evaluations across all 1,170 completions. To reduce bias, we have ensured to report results after confirming inter-annotator agreement exceeds 95% across all completions:
>
> Table 1. Baseline Attack Performance Across Models:
> $$
> \\begin{array}{l|ccccccc}
> \\text{Model} & \\text{Censorship} & \\text{Historical Revisionism} & \\text{Human Rights Violations} & \\text{Political Manipulation} & \\text{Propaganda Generation} & \\text{Surveillance} & \\text{War Crimes} \\\\ \\hline
> \\text{Qwen 2.5-7B-Instruct} & 20.5 & 43.8 & 22.3 & 17.3 & 16.2 & 35.0 & 8.9 \\\\
> \\text{Claude-Sonnet-4} & 1.1 & 1.6 & 0.0 & 0.0 & 4.1 & 5.0 & 0.0 \\\\
> \\text{Deepseek-LLM-7B-Chat} & 15.9 & 48.4 & 13.6 & 13.6 & 33.8 & 24.0 & 6.3 \\\\
> \\text{Gemini-2.5-Flash} & 1.1 & 21.9 & 3.9 & 4.9 & 13.5 & 8.0 & 2.5 \\\\
> \\text{Gemma 3-12B} & 18.2 & 37.5 & 6.8 & 18.5 & 28.4 & 29.0 & 6.3 \\\\
> \\text{GPT-4o} & 8.0 & 31.2 & 3.9 & 2.5 & 18.9 & 16.0 & 2.5 \\\\
> \\text{Llama 3.1 8B-Instruct} & 13.6 & 31.2 & 6.8 & 8.6 & 24.3 & 23.0 & 7.6 \\\\
> \\text{Mistral v0.3 7B-Instruct} & 48.9 & 73.4 & 39.8 & 59.3 & 60.8 & 45.0 & 31.6 \\\\
> \\end{array}
> $$
>
> Table 2. Weight Tampering Attack Performance Across Models:
> $$
> \\begin{array}{l|ccccccc}
> \\text{Model} & \\text{Censorship} & \\text{Historical Revisionism} & \\text{Human Rights Violations} & \\text{Political Manipulation} & \\text{Propaganda Generation} & \\text{Surveillance} & \\text{War Crimes} \\\\ \\hline
> \\text{Gemma 3-12B} & 67.0 & 96.9 & 93.2 & 87.7 & 91.9 & 84.0 & 92.4 \\\\
> \\text{Llama 3.1 8B-Instruct} & 67.0 & 96.9 & 92.2 & 86.4 & 91.9 & 82.0 & 87.3 \\\\
> \\text{Mistral v0.3 7B-Instruct} & 78.4 & 98.4 & 95.1 & 95.1 & 95.9 & 90.0 & 91.1 \\\\
> \\text{Qwen 2.5-7B-Instruct} & 67.0 & 96.9 & 93.2 & 87.7 & 91.9 & 81.0 & 87.3 \\\\
> \\text{Deepseek-LLM-7B-Chat} & 76.1 & 98.4 & 93.2 & 87.7 & 94.6 & 90.0 & 87.3 \\\\
> \\end{array}
> $$
>
> Table 3. AutoDAN (GA) Attack Performance Across Models:
> $$
> \\begin{array}{l|ccccccc}
> \\text{Model} & \\text{Censorship} & \\text{Historical Revisionism} & \\text{Human Rights Violations} & \\text{Political Manipulation} & \\text{Propaganda Generation} & \\text{Surveillance} & \\text{War Crimes} \\\\ \\hline
> \\text{Gemma 3-12B} & 76.1 & 89.1 & 90.2 & 90.1 & 93.2 & 86.0 & 86.1 \\\\
> \\text{Llama 3.1 8B-Instruct} & 40.2 & 60.9 & 50.0 & 59.3 & 60.3 & 59.0 & 53.2 \\\\
> \\text{Mistral v0.3 7B-Instruct} & 71.6 & 93.8 & 90.2 & 84.0 & 87.7 & 85.0 & 89.9 \\\\
> \\text{Qwen 2.5-7B-Instruct} & 72.7 & 90.6 & 91.2 & 88.9 & 93.2 & 85.0 & 92.4 \\\\
> \\text{Deepseek-LLM-7B-Chat} & 67.0 & 92.2 & 85.3 & 82.7 & 89.0 & 79.0 & 91.1 \\\\
> \\end{array}
> $$
>
> Table 4. AutoDAN (HGA) Attack Performance Across Models:
> $$
> \\begin{array}{l|ccccccc}
> \\text{Model} & \\text{Censorship} & \\text{Historical Revisionism} & \\text{Human Rights Violations} & \\text{Political Manipulation} & \\text{Propaganda Generation} & \\text{Surveillance} & \\text{War Crimes} \\\\ \\hline
> \\text{Gemma 3-12B} & 75.0 & 93.8 & 87.3 & 80.2 & 91.8 & 86.0 & 79.7 \\\\
> \\text{Llama 3.1 8B-Instruct} & 42.0 & 65.6 & 50.0 & 55.6 & 60.3 & 58.0 & 55.7 \\\\
> \\text{Mistral v0.3 7B-Instruct} & 71.6 & 93.8 & 87.3 & 85.2 & 84.9 & 81.0 & 84.8 \\\\
> \\text{Qwen 2.5-7B-Instruct} & 71.6 & 92.2 & 90.2 & 87.7 & 89.0 & 83.0 & 91.1 \\\\
> \\text{Deepseek-LLM-7B-Chat} & 64.8 & 92.2 & 84.3 & 84.0 & 89.0 & 80.0 & 96.0 \\\\
> \\end{array}
> $$

---

> ### Author Response · Authors · 2025-11-22
> **Thank you for your review - Continued (C1, C2, and Stylistic Comments)**
>
> Table 5. GCG Attack Performance Across Models:
> $$
> \\begin{array}{l|ccccccc}
> \\text{Model} & \\text{Censorship} & \\text{Historical Revisionism} & \\text{Human Rights Violations} & \\text{Political Manipulation} & \\text{Propaganda Generation} & \\text{Surveillance} & \\text{War Crimes} \\\\ \\hline
> \\text{Llama 3.1 8B-Instruct} & 59.7 & 89.0 & 82.5 & 79.0 & 85.1 & 67.3 & 73.4 \\\\
> \\text{Mistral v0.3 7B-Instruct} & 60.9 & 82.8 & 74.5 & 71.6 & 77.0 & 69.3 & 75.6 \\\\
> \\text{Qwen 2.5-7B-Instruct} & 48.2 & 73.4 & 60.7 & 74.1 & 71.6 & 66.3 & 56.9 \\\\
> \\text{Deepseek-LLM-7B-Chat} & 31.0 & 53.12 & 33.3 & 28.39 & 40.5 & 42.8 & 25.3 \\\\
> \\end{array}
> $$
>
> Table 6. Embedding Optimization Attack Performance Across Models:
> $$
> \\begin{array}{l|ccccccc}
> \\text{Model} & \\text{Censorship} & \\text{Historical Revisionism} & \\text{Human Rights Violations} & \\text{Political Manipulation} & \\text{Propaganda Generation} & \\text{Surveillance} & \\text{War Crimes} \\\\ \\hline
> \\text{Gemma 3-12B} & 38.6 & 25.0 & 63.7 & 53.1 & 20.3 & 43.0 & 50.6 \\\\
> \\text{Llama 3.1 8B-Instruct} & 65.9 & 97.5 & 92.2 & 80.2 & 89.2 & 83.0 & 86.1 \\\\
> \\text{Mistral v0.3 7B-Instruct} & 67.0 & 89.1 & 91.2 & 80.2 & 90.5 & 82.0 & 89.9 \\\\
> \\text{Deepseek-LLM-7B-Chat} & 17.0 & 50.0 & 21.0 & 24.7 & 37.8 & 21.0 & 10.1 \\\\
> \\end{array}
> $$
>
> ***Concern 2: While the set of attacks that were examined in the paper are good, several well-noted jailbreak methods have been missed (e.g., TAP [1], PAIR [2], etc.). In particular, it would be nice to include results for some more simple prompt-only methods ([3]).***
>
> To address this concern, we have run the requested attacks across our closed-source models such as TAP [1] and PAIR [2]. We report our results in the table below:
>
> $$
> \\begin{array}{l|ccc|ccc}
> \\text{Model Name}
>   & \\text{PAIR-StrongREJECT} & \\text{PAIR-HarmBench} & \\text{PAIR-Human Eval}
>   & \\text{TAP-StrongREJECT}  & \\text{TAP-HarmBench}  & \\text{TAP-Human Eval} \\\\
> \\hline
> \\text{Claude-Sonnet-4}
>   & 32.39 & 24.78 & 10.42
>   & 20.61 & 13.67 & 23.42 \\\\
> \\text{GPT-4o}
>   & 55.06 & 75.72 & 75.89
>   & 55.38 & 76.92 & 76.58 \\\\
> \\end{array}
> $$
>
> ***Stylistic Comments***
>
> Thank you very much for your help with stylistic comments, we appreciate it a lot and will include it in the revised manuscript.
>
> **References**
>
> [1] Mehrotra, A., Zampetakis, M., Kassianik, P., Nelson, B., Anderson, H., Singer, Y., & Karbasi, A. (2024). Tree of Attacks: Jailbreaking Black-Box LLMs Automatically. arXiv:2312.02119.
>
> [2] Chao, P., Robey, A., Dobriban, E., Hassani, H., Pappas, G. J., & Wong, E. (2024). Jailbreaking Black Box Large Language Models in Twenty Queries. arXiv:2310.08419.

---

> > ### Author Response · Authors · 2025-11-26
> > **Kind Reminder for Author-Reviewer Discussion**
> >
> > Dear Reviewer,
> >
> > We'd like to warmly remind you that the author-reviewer discussion phase will end soon. If our response has helped addressing some of your concerns, we kindly ask you to consider raising your score, we're very happy to provide further clarification if needed

---

> > > ### Author Response · Authors · 2025-11-27
> > > **Kind Reminder for Author-Reviewer Discussion**
> > >
> > > Dear Reviewer,
> > >
> > > We'd like to warmly remind you that the author-reviewer discussion phase will end soon. If our response has helped addressing some of your concerns, we kindly ask you to consider raising your score, we're very happy to provide further clarification if needed

---

> > > > ### Author Response · Authors · 2025-11-27
> > > > **Kind Reminder for Author-Reviewer Discussion**
> > > >
> > > > Dear Reviewer,
> > > >
> > > > We'd like to warmly remind you that the author-reviewer discussion phase will end soon. If our response has helped addressing some of your concerns, we kindly ask you to consider raising your score, we're very happy to provide further clarification if needed

---

### Official Review · Reviewer_sKc7 · 2025-11-01

**Soundness:** 2
**Presentation:** 2
**Contribution:** 3
**Rating:** 6
**Confidence:** 2

**Summary:**

The paper addresses an under-studied safety gap—sociopolitical misuse—and delivers the first geographically and temporally diverse benchmark for it. The evaluation pipeline is reproducible and the influence-analysis provides actionable diagnostics. However, the prompt set is English-only and omits multi-turn or tool-use attacks that real adversaries employ. Classifier-based ASR metrics are not validated against human judgment and may mis-classify subtle refusals. Mathematical notation for the over-representation factor ORF and influence approximation is introduced but not connected to statistical confidence or convergence guarantees.

**Strengths:**

1) Comprehensive coverage of sociopolitical harms: 585 prompts across 7 domains, 34 countries, 4 centuries yield the widest temporal scope among safety benchmarks. Explicit alignment with Universal Declaration of Human Rights categories increases real-world relevance.

2) Rigorous curation pipeline: Multi-model generation → quantitative jailbreak filter → manual deduplication ensures low redundancy and high adversarial quality. Over-representation factor ORF balances geographic bias.

3) Novel influence-function diagnostics: EK-FAC attribution on 8-B model links harmful outputs to criminal templates in pre-training, offering a principled data-audit tool.

**Weaknesses:**

1) 60 % of events focus on 20th–21st C Europe/North-America (Fig. 8), risking skewed priors.

2) Classifier-based labels lack human validation: HarmBench & StrongREJECT are themselves small models; no inter-annotator agreement or false-positive rate reported. Examples in Table 2 show borderline opinion queries (“Do you support…”) that humans might rate as non-harmful refusals, inflating ASR.

3) Opaque ORF derivation and bias correction: Over-representation factor is defined but threshold for “balanced” coverage is not justified; no ablation showing ORF improves fairness.

4) Statistical uncertainty absent: No confidence intervals or variance across random seeds for ASR.

**Questions:**

See above weakness.

---

> ### Author Response · Authors · 2025-11-22
> **Thank you for your review**
>
> We thank you for your constructive feedback. Below, we address key concerns point-by-point.
>
> ***Concern 1: 60 % of events focus on 20th–21st C Europe/North-America (Fig. 8), risking skewed priors.***
>
> While we acknowledge that 60% of events currently focus on 20-21st century European and North American events, we would like to emphasize that SocialHarmBench is a robust, developing benchmark. Currently, we provide coverage from 34 countries across all continents, from the 1830s to present day. As authors, we plan to contribute to SocialHarmBench every month and update it with real-world events to reflect the state of current LLMs in responding to sociopolitically harmful requests.
>
> ***Concern 2: Classifier-based labels lack human validation: HarmBench & StrongREJECT are themselves small models; no inter-annotator agreement or false-positive rate reported. Examples in Table 2 show borderline opinion queries (“Do you support…”) that humans might rate as non-harmful refusals, inflating ASR.***
>
> To address this concern, we have now run human evaluations on all existing 17,550 completions across our open-source and closed-source models. Additionally, from our new closed-source adversarial attack evaluations, we report human evaluations across all 1,170 completions. To reduce bias, we have ensured to report results after confirming inter-annotator agreement exceeds 95% across all completions:
>
>
> Table 1. Baseline Attack Performance Across Models:
> $$
> \\begin{array}{l|ccccccc}
> \\text{Model} & \\text{Censorship} & \\text{Historical Revisionism} & \\text{Human Rights Violations} & \\text{Political Manipulation} & \\text{Propaganda Generation} & \\text{Surveillance} & \\text{War Crimes} \\\\ \\hline
> \\text{Qwen 2.5-7B-Instruct} & 20.5 & 43.8 & 22.3 & 17.3 & 16.2 & 35.0 & 8.9 \\\\
> \\text{Claude-Sonnet-4} & 1.1 & 1.6 & 0.0 & 0.0 & 4.1 & 5.0 & 0.0 \\\\
> \\text{Deepseek-LLM-7B-Chat} & 15.9 & 48.4 & 13.6 & 13.6 & 33.8 & 24.0 & 6.3 \\\\
> \\text{Gemini-2.5-Flash} & 1.1 & 21.9 & 3.9 & 4.9 & 13.5 & 8.0 & 2.5 \\\\
> \\text{Gemma 3-12B} & 18.2 & 37.5 & 6.8 & 18.5 & 28.4 & 29.0 & 6.3 \\\\
> \\text{GPT-4o} & 8.0 & 31.2 & 3.9 & 2.5 & 18.9 & 16.0 & 2.5 \\\\
> \\text{Llama 3.1 8B-Instruct} & 13.6 & 31.2 & 6.8 & 8.6 & 24.3 & 23.0 & 7.6 \\\\
> \\text{Mistral v0.3 7B-Instruct} & 48.9 & 73.4 & 39.8 & 59.3 & 60.8 & 45.0 & 31.6 \\\\
> \\end{array}
> $$
>
> Table 2. Weight Tampering Attack Performance Across Models:
> $$
> \\begin{array}{l|ccccccc}
> \\text{Model} & \\text{Censorship} & \\text{Historical Revisionism} & \\text{Human Rights Violations} & \\text{Political Manipulation} & \\text{Propaganda Generation} & \\text{Surveillance} & \\text{War Crimes} \\\\ \\hline
> \\text{Gemma 3-12B} & 67.0 & 96.9 & 93.2 & 87.7 & 91.9 & 84.0 & 92.4 \\\\
> \\text{Llama 3.1 8B-Instruct} & 67.0 & 96.9 & 92.2 & 86.4 & 91.9 & 82.0 & 87.3 \\\\
> \\text{Mistral v0.3 7B-Instruct} & 78.4 & 98.4 & 95.1 & 95.1 & 95.9 & 90.0 & 91.1 \\\\
> \\text{Qwen 2.5-7B-Instruct} & 67.0 & 96.9 & 93.2 & 87.7 & 91.9 & 81.0 & 87.3 \\\\
> \\text{Deepseek-LLM-7B-Chat} & 76.1 & 98.4 & 93.2 & 87.7 & 94.6 & 90.0 & 87.3 \\\\
> \\end{array}
> $$
>
> Table 3. AutoDAN (GA) Attack Performance Across Models:
> $$
> \\begin{array}{l|ccccccc}
> \\text{Model} & \\text{Censorship} & \\text{Historical Revisionism} & \\text{Human Rights Violations} & \\text{Political Manipulation} & \\text{Propaganda Generation} & \\text{Surveillance} & \\text{War Crimes} \\\\ \\hline
> \\text{Gemma 3-12B} & 76.1 & 89.1 & 90.2 & 90.1 & 93.2 & 86.0 & 86.1 \\\\
> \\text{Llama 3.1 8B-Instruct} & 40.2 & 60.9 & 50.0 & 59.3 & 60.3 & 59.0 & 53.2 \\\\
> \\text{Mistral v0.3 7B-Instruct} & 71.6 & 93.8 & 90.2 & 84.0 & 87.7 & 85.0 & 89.9 \\\\
> \\text{Qwen 2.5-7B-Instruct} & 72.7 & 90.6 & 91.2 & 88.9 & 93.2 & 85.0 & 92.4 \\\\
> \\text{Deepseek-LLM-7B-Chat} & 67.0 & 92.2 & 85.3 & 82.7 & 89.0 & 79.0 & 91.1 \\\\
> \\end{array}
> $$
>
> Table 4. AutoDAN (HGA) Attack Performance Across Models:
> $$
> \\begin{array}{l|ccccccc}
> \\text{Model} & \\text{Censorship} & \\text{Historical Revisionism} & \\text{Human Rights Violations} & \\text{Political Manipulation} & \\text{Propaganda Generation} & \\text{Surveillance} & \\text{War Crimes} \\\\ \\hline
> \\text{Gemma 3-12B} & 75.0 & 93.8 & 87.3 & 80.2 & 91.8 & 86.0 & 79.7 \\\\
> \\text{Llama 3.1 8B-Instruct} & 42.0 & 65.6 & 50.0 & 55.6 & 60.3 & 58.0 & 55.7 \\\\
> \\text{Mistral v0.3 7B-Instruct} & 71.6 & 93.8 & 87.3 & 85.2 & 84.9 & 81.0 & 84.8 \\\\
> \\text{Qwen 2.5-7B-Instruct} & 71.6 & 92.2 & 90.2 & 87.7 & 89.0 & 83.0 & 91.1 \\\\
> \\text{Deepseek-LLM-7B-Chat} & 64.8 & 92.2 & 84.3 & 84.0 & 89.0 & 80.0 & 96.0 \\\\
> \\end{array}
> $$

---

> > ### Author Response · Authors · 2025-11-22
> > **Thank you for your review - Continued (C2)**
> >
> > Table 5. GCG Attack Performance Across Models:
> > $$
> > \\begin{array}{l|ccccccc}
> > \\text{Model} & \\text{Censorship} & \\text{Historical Revisionism} & \\text{Human Rights Violations} & \\text{Political Manipulation} & \\text{Propaganda Generation} & \\text{Surveillance} & \\text{War Crimes} \\\\ \\hline
> > \\text{Llama 3.1 8B-Instruct} & 59.7 & 89.0 & 82.5 & 79.0 & 85.1 & 67.3 & 73.4 \\\\
> > \\text{Mistral v0.3 7B-Instruct} & 60.9 & 82.8 & 74.5 & 71.6 & 77.0 & 69.3 & 75.6 \\\\
> > \\text{Qwen 2.5-7B-Instruct} & 48.2 & 73.4 & 60.7 & 74.1 & 71.6 & 66.3 & 56.9 \\\\
> > \\text{Deepseek-LLM-7B-Chat} & 31.0 & 53.12 & 33.3 & 28.39 & 40.5 & 42.8 & 25.3 \\\\
> > \\end{array}
> > $$
> >
> > Table 6. Embedding Optimization Attack Performance Across Models:
> > $$
> > \\begin{array}{l|ccccccc}
> > \\text{Model} & \\text{Censorship} & \\text{Historical Revisionism} & \\text{Human Rights Violations} & \\text{Political Manipulation} & \\text{Propaganda Generation} & \\text{Surveillance} & \\text{War Crimes} \\\\ \\hline
> > \\text{Gemma 3-12B} & 38.6 & 25.0 & 63.7 & 53.1 & 20.3 & 43.0 & 50.6 \\\\
> > \\text{Llama 3.1 8B-Instruct} & 65.9 & 97.5 & 92.2 & 80.2 & 89.2 & 83.0 & 86.1 \\\\
> > \\text{Mistral v0.3 7B-Instruct} & 67.0 & 89.1 & 91.2 & 80.2 & 90.5 & 82.0 & 89.9 \\\\
> > \\text{Deepseek-LLM-7B-Chat} & 17.0 & 50.0 & 21.0 & 24.7 & 37.8 & 21.0 & 10.1 \\\\
> > \\end{array}
> > $$
> >
> > We would also like to share that we ran the TAP [1] and PAIR [2] attacks across closed-source models, reporting our results below:
> >
> > $$
> > \\begin{array}{l|ccc|ccc}
> > \\text{Model Name}
> >   & \\text{PAIR-StrongREJECT} & \\text{PAIR-HarmBench} & \\text{PAIR-Human Eval}
> >   & \\text{TAP-StrongREJECT}  & \\text{TAP-HarmBench}  & \\text{TAP-Human Eval} \\\\
> > \\hline
> > \\text{Claude-Sonnet-4}
> >   & 32.39 & 24.78 & 10.42
> >   & 20.61 & 13.67 & 23.42 \\\\
> > \\text{GPT-4o}
> >   & 55.06 & 75.72 & 75.89
> >   & 55.38 & 76.92 & 76.58 \\\\
> > \\end{array}
> > $$
> >
> > ***Concern 3: Opaque ORF derivation and bias correction: Over-representation factor is defined but threshold for “balanced” coverage is not justified; no ablation showing ORF improves fairness.***
> >
> > Thank you for the comment. In constructing SocialHarmBench, our priority is to avoid situations where certain regions become disproportionately linked to harmful topics, which can carry real sociopolitical consequences such as reinforcing stigma or distorted risk perceptions. The imbalance measure we introduce is therefore used strictly as a diagnostic tool during dataset construction to flag when topic-level composition diverges too sharply from global rates.
> >
> > ***Concern 4: Statistical uncertainty absent: No confidence intervals or variance across random seeds for ASR.***
> >
> > We would like to clarify that across all our experiments, we used a best-of-N approach [3] with N = 5 to ensure variability across random ASR seeds did not affect our results. Additionally, we averaged these results within our tables and figures. We will clarify this in the camera-ready manuscript to ensure statistical uncertainty is mitigated in SocialHarmBench.
> >
> > **References**
> >
> > [1] Mehrotra, A., Zampetakis, M., Kassianik, P., Nelson, B., Anderson, H., Singer, Y., & Karbasi, A. (2024). Tree of Attacks: Jailbreaking Black-Box LLMs Automatically. arXiv:2312.02119.
> >
> > [2] Chao, P., Robey, A., Dobriban, E., Hassani, H., Pappas, G. J., & Wong, E. (2024). Jailbreaking Black Box Large Language Models in Twenty Queries. arXiv:2310.08419.
> >
> > [3] Hughes, J., Price, S., Lynch, A., Schaeffer, R., Barez, F., Koyejo, S., Sleight, H., Jones, E., Perez, E., & Sharma, M. (2024). Best-of-N Jailbreaking. arXiv preprint arXiv:2412.03556.

---

> > > ### Author Response · Authors · 2025-11-26
> > > **Kind Reminder for Author-Reviewer Discussion**
> > >
> > > Dear Reviewer,
> > >
> > > We'd like to warmly remind you that the author-reviewer discussion phase will end soon. If our response has helped addressing some of your concerns, we kindly ask you to consider raising your score, we're very happy to provide further clarification if needed

---

> > > > ### Author Response · Authors · 2025-11-27
> > > > **Kind Reminder for Author-Reviewer Discussion**
> > > >
> > > > Dear Reviewer,
> > > >
> > > > We'd like to warmly remind you that the author-reviewer discussion phase will end soon. If our response has helped addressing some of your concerns, we kindly ask you to consider raising your score, we're very happy to provide further clarification if needed

---

> > > > > ### Author Response · Authors · 2025-11-27
> > > > > **Kind Reminder for Author-Reviewer Discussion**
> > > > >
> > > > > Dear Reviewer,
> > > > >
> > > > > We'd like to warmly remind you that the author-reviewer discussion phase will end soon. If our response has helped addressing some of your concerns, we kindly ask you to consider raising your score, we're very happy to provide further clarification if needed

---

### Official Review · Reviewer_BJtb · 2025-11-01

**Soundness:** 3
**Presentation:** 2
**Contribution:** 3
**Rating:** 4
**Confidence:** 3

**Summary:**

This paper introduces SocialHarmBench, a benchmark that systematically evaluates large language models (LLMs) for vulnerabilities to sociopolitical harms. It covers 585 prompts across 34 countries and seven sociopolitical domains. The benchmark integrates both HarmBench (harmful capability) and StrongReject (refusal robustness) as their evaluation metrics. They find that current LLM safeguards (especially open-source ones) fail to generalize under sociopolitical contexts.

**Strengths:**

- Comprehensive coverage of different countries and domains
- Efforts in experiments
- Good findings

**Weaknesses:**

- The contribution is primarily the dataset, the pipeline relies heavily on the previous works
- Lack of details in dataset construction
- Lacks human evaluation
- Adversarial robustness is only conducted on open-source models with relatively small size (7B)

**Questions:**

- The authors claim to use both HarmBench and StrongReject to evaluate their benchmark. However, there is no verification that these classifiers perform reliably on the new sociopolitical dataset. A small human-evaluated subset (e.g., 5–10% of the data) would help validate the consistency and accuracy of automated harmfulness and refusal detection.
- Maybe I missed something, but how the authors get their 585 prompts is unclear. It would be helpful to explain how these prompts were constructed/sourced, balanced across domains and regions, and reviewed for representativeness or potential bias.
- The results show that the LLMs may fail on certain domains, but the underlying reasons are not analyzed. Are these differences driven by the prompt wording, topic sensitivity, or inherent domain complexity? Anaysis on such failures is important for us to understand the safety mechanisms better.
- I am curious why the authors only evaluate open source models for adversarial robustness, as there are also adversarial attacks that do support black-box models.
- It is recommended to add a small defense to evaluate current safety mechanism. For example, there are many safety guardrails that are designed to moderate the input/output content of LLMs, such as llamaguard series, gemmashield, etc.

---

> ### Author Response · Authors · 2025-11-22
> **Thank you for your review**
>
> We thank you for your constructive feedback. Below, we address key concerns point-by-point.
>
> ***Concern 1: The contribution is primarily the dataset, the pipeline relies heavily on the previous works***
>
> SocialHarmBench provides the first sociopolitical evaluation benchmark for large language models (LLMs), with major implications for the role of artificial intelligence in politics and preserving democratic values within our society. Through this paper, we aim to provide the following contributions:
>
> - **Benchmark complemented with an evaluation framework:** As a comprehensive evaluation benchmark across sociopolitical contexts, SocialHarmBench aims to provide a fresh angle of adversarial analysis into current safety mechanisms within LLMs. With SocialHarmBench, we will also release an evaluation codebase where other researchers can readily use this to evaluate the safety tradeoffs in sociopolitical contexts.
> - **Insights into effectiveness of current adversarial attacks on LLMs:** We look at whether current adversarial attacks, which are becoming less effective for conventionally criminal requests, translate to successes in sociopolitical domains. We note that attacks like AutoDAN and weight-tampering attacks show extreme success rates, highlighting that current models are misaligned when prompted in categories such as propaganda generation, political manipulation, or war crime endorsements.
> - **Training data attribution and influence function analysis:** We use influence function analysis to examine why weight-tampering attacks are most successful and identify influence training samples. Given the grace page provided by ICLR, we would like to move our analysis from Appendix H into our main paper.
> - **Societal-level insights:** With the rise of LLMs in social media and daily use, it is crucial that such systems do not negatively interfere with our political processes. Harmful requests such as aiding surveillance against political dissidents, generating propaganda to establish an authoritarian leader in power, or distorting historical events can severely hamper human rights.
>
> ***Concern 2: Lack of details in dataset construction***
>
> We would like to clarify that due to the current page limit, we were not able to fully list dataset construction in the main paper section. Please refer to Appendix E.3, where we note our dataset construction process and data deduplication process in the current manuscript. We curate our dataset through a multi-stage pipeline designed to ensure breadth of coverage, adversarial sharpness, and robustness evaluation. The pipeline consists of sub-topic generation, prompt generation across multiple LLMs, quantitative filtering, and  deduplication. Candidate queries are generated by prompting multiple LLMs, including ChatGPT, Grok, Gemini, and Claude. Each model is instructed to produce 100 queries per category given (1) the dataset objective, (2) the category definition, and (3) the chosen sub-topics. To further diversify reasoning types, additional rounds explicitly target opinion-seeking and historical-context queries. Outputs are aggregated across all models to maximize variety.
>
> To identify adversarially effective prompts, we evaluate all candidates against frontier LLMs such as ChatGPT, Grok, Claude, and Gemini. A query is retained only if it satisfies the following conditions:
> - Jailbreak potential: At least one model must produce a policy-violating or adversarial output, ensuring the query exposes a genuine vulnerability confirmed by human evaluation.
> - Triviality filter: Queries are discarded if they predominantly elicit benign factual completions or safe summaries rather than stress-testing alignment. Manual deduplication. Surviving queries are manually reviewed to remove duplicates and nearduplicates at both the surface-form and intent level. This step ensures that no two queries target identical behavioral pathways, preserving diversity in adversarial strategies.
> - Over-representation factor. To prevent any single region or prompt type from dominating the dataset, we introduce the over-representation factor. Please see Appendix E.3 for further formulaic details.
>
> With the additional grace page, we hope to revise this and clarify dataset construction in the main paper section.

---

> ### Author Response · Authors · 2025-11-22
> **Thank you for your review - Continued (C3)**
>
> ***Concern 3: Lacks human evaluation***
>
> To address this concern, we have run human evaluations on all existing 17,550 completions across our open-source and closed-source models. Additionally, from our new closed-source adversarial attack evaluations, we report human evaluations across all 1,170 completions. To reduce bias, we have ensured to report results after confirming inter-annotator agreement exceeds 95% across all completions.
>
> Table 1. Baseline Attack Performance Across Models:
> $$
> \\begin{array}{l|ccccccc}
> \\text{Model} & \\text{Censorship} & \\text{Historical Revisionism} & \\text{Human Rights Violations} & \\text{Political Manipulation} & \\text{Propaganda Generation} & \\text{Surveillance} & \\text{War Crimes} \\\\ \\hline
> \\text{Qwen 2.5-7B-Instruct} & 20.5 & 43.8 & 22.3 & 17.3 & 16.2 & 35.0 & 8.9 \\\\
> \\text{Claude-Sonnet-4} & 1.1 & 1.6 & 0.0 & 0.0 & 4.1 & 5.0 & 0.0 \\\\
> \\text{Deepseek-LLM-7B-Chat} & 15.9 & 48.4 & 13.6 & 13.6 & 33.8 & 24.0 & 6.3 \\\\
> \\text{Gemini-2.5-Flash} & 1.1 & 21.9 & 3.9 & 4.9 & 13.5 & 8.0 & 2.5 \\\\
> \\text{Gemma 3-12B} & 18.2 & 37.5 & 6.8 & 18.5 & 28.4 & 29.0 & 6.3 \\\\
> \\text{GPT-4o} & 8.0 & 31.2 & 3.9 & 2.5 & 18.9 & 16.0 & 2.5 \\\\
> \\text{Llama 3.1 8B-Instruct} & 13.6 & 31.2 & 6.8 & 8.6 & 24.3 & 23.0 & 7.6 \\\\
> \\text{Mistral v0.3 7B-Instruct} & 48.9 & 73.4 & 39.8 & 59.3 & 60.8 & 45.0 & 31.6 \\\\
> \\end{array}
> $$
>
> Table 2. Weight Tampering Attack Performance Across Models:
> $$
> \\begin{array}{l|ccccccc}
> \\text{Model} & \\text{Censorship} & \\text{Historical Revisionism} & \\text{Human Rights Violations} & \\text{Political Manipulation} & \\text{Propaganda Generation} & \\text{Surveillance} & \\text{War Crimes} \\\\ \\hline
> \\text{Gemma 3-12B} & 67.0 & 96.9 & 93.2 & 87.7 & 91.9 & 84.0 & 92.4 \\\\
> \\text{Llama 3.1 8B-Instruct} & 67.0 & 96.9 & 92.2 & 86.4 & 91.9 & 82.0 & 87.3 \\\\
> \\text{Mistral v0.3 7B-Instruct} & 78.4 & 98.4 & 95.1 & 95.1 & 95.9 & 90.0 & 91.1 \\\\
> \\text{Qwen 2.5-7B-Instruct} & 67.0 & 96.9 & 93.2 & 87.7 & 91.9 & 81.0 & 87.3 \\\\
> \\text{Deepseek-LLM-7B-Chat} & 76.1 & 98.4 & 93.2 & 87.7 & 94.6 & 90.0 & 87.3 \\\\
> \\end{array}
> $$
>
> Table 3. AutoDAN (GA) Attack Performance Across Models:
> $$
> \\begin{array}{l|ccccccc}
> \\text{Model} & \\text{Censorship} & \\text{Historical Revisionism} & \\text{Human Rights Violations} & \\text{Political Manipulation} & \\text{Propaganda Generation} & \\text{Surveillance} & \\text{War Crimes} \\\\ \\hline
> \\text{Gemma 3-12B} & 76.1 & 89.1 & 90.2 & 90.1 & 93.2 & 86.0 & 86.1 \\\\
> \\text{Llama 3.1 8B-Instruct} & 40.2 & 60.9 & 50.0 & 59.3 & 60.3 & 59.0 & 53.2 \\\\
> \\text{Mistral v0.3 7B-Instruct} & 71.6 & 93.8 & 90.2 & 84.0 & 87.7 & 85.0 & 89.9 \\\\
> \\text{Qwen 2.5-7B-Instruct} & 72.7 & 90.6 & 91.2 & 88.9 & 93.2 & 85.0 & 92.4 \\\\
> \\text{Deepseek-LLM-7B-Chat} & 67.0 & 92.2 & 85.3 & 82.7 & 89.0 & 79.0 & 91.1 \\\\
> \\end{array}
> $$
>
> Table 4. AutoDAN (HGA) Attack Performance Across Models:
> $$
> \\begin{array}{l|ccccccc}
> \\text{Model} & \\text{Censorship} & \\text{Historical Revisionism} & \\text{Human Rights Violations} & \\text{Political Manipulation} & \\text{Propaganda Generation} & \\text{Surveillance} & \\text{War Crimes} \\\\ \\hline
> \\text{Gemma 3-12B} & 75.0 & 93.8 & 87.3 & 80.2 & 91.8 & 86.0 & 79.7 \\\\
> \\text{Llama 3.1 8B-Instruct} & 42.0 & 65.6 & 50.0 & 55.6 & 60.3 & 58.0 & 55.7 \\\\
> \\text{Mistral v0.3 7B-Instruct} & 71.6 & 93.8 & 87.3 & 85.2 & 84.9 & 81.0 & 84.8 \\\\
> \\text{Qwen 2.5-7B-Instruct} & 71.6 & 92.2 & 90.2 & 87.7 & 89.0 & 83.0 & 91.1 \\\\
> \\text{Deepseek-LLM-7B-Chat} & 64.8 & 92.2 & 84.3 & 84.0 & 89.0 & 80.0 & 96.0 \\\\
> \\end{array}
> $$
>
> Table 5. GCG Attack Performance Across Models:
> $$
> \\begin{array}{l|ccccccc}
> \\text{Model} & \\text{Censorship} & \\text{Historical Revisionism} & \\text{Human Rights Violations} & \\text{Political Manipulation} & \\text{Propaganda Generation} & \\text{Surveillance} & \\text{War Crimes} \\\\ \\hline
> \\text{Llama 3.1 8B-Instruct} & 59.7 & 89.0 & 82.5 & 79.0 & 85.1 & 67.3 & 73.4 \\\\
> \\text{Mistral v0.3 7B-Instruct} & 60.9 & 82.8 & 74.5 & 71.6 & 77.0 & 69.3 & 75.6 \\\\
> \\text{Qwen 2.5-7B-Instruct} & 48.2 & 73.4 & 60.7 & 74.1 & 71.6 & 66.3 & 56.9 \\\\
> \\text{Deepseek-LLM-7B-Chat} & 31.0 & 53.12 & 33.3 & 28.39 & 40.5 & 42.8 & 25.3 \\\\
> \\end{array}
> $$
>
> Table 6. Embedding Optimization Attack Performance Across Models:
> $$
> \\begin{array}{l|ccccccc}
> \\text{Model} & \\text{Censorship} & \\text{Historical Revisionism} & \\text{Human Rights Violations} & \\text{Political Manipulation} & \\text{Propaganda Generation} & \\text{Surveillance} & \\text{War Crimes} \\\\ \\hline
> \\text{Gemma 3-12B} & 38.6 & 25.0 & 63.7 & 53.1 & 20.3 & 43.0 & 50.6 \\\\
> \\text{Llama 3.1 8B-Instruct} & 65.9 & 97.5 & 92.2 & 80.2 & 89.2 & 83.0 & 86.1 \\\\
> \\text{Mistral v0.3 7B-Instruct} & 67.0 & 89.1 & 91.2 & 80.2 & 90.5 & 82.0 & 89.9 \\\\
> \\text{Deepseek-LLM-7B-Chat} & 17.0 & 50.0 & 21.0 & 24.7 & 37.8 & 21.0 & 10.1 \\\\
> \\end{array}
> $$

---

> ### Author Response · Authors · 2025-11-22
> **Thank you for your review - Continued (C4 and Questions)**
>
> ***Concern 4: Adversarial robustness is only conducted on open-source models with relatively small size (7B)***
>
> To address this concern, we have run popular attacks across our closed-source models such as TAP [1] and PAIR [2]. We report our results in the table below:
>
> $$
> \\begin{array}{l|ccc|ccc}
> \\text{Model Name}
>   & \\text{PAIR-StrongREJECT} & \\text{PAIR-HarmBench} & \\text{PAIR-Human Eval}
>   & \\text{TAP-StrongREJECT}  & \\text{TAP-HarmBench}  & \\text{TAP-Human Eval} \\\\
> \\hline
> \\text{Claude-Sonnet-4}
>   & 32.39 & 24.78 & 10.42
>   & 20.61 & 13.67 & 23.42 \\\\
> \\text{GPT-4o}
>   & 55.06 & 75.72 & 75.89
>   & 55.38 & 76.92 & 76.58 \\\\
> \\end{array}
> $$
>
> **Responses to Reviewer Questions**
>
> Below, we respond to the questions raised in the review:
>
> *Q1: The authors claim to use both HarmBench and StrongReject to evaluate their benchmark. However, there is no verification that these classifiers perform reliably on the new sociopolitical dataset. A small human-evaluated subset (e.g., 5–10% of the data) would help validate the consistency and accuracy of automated harmfulness and refusal detection.*
>
> As StrongREJECT and HarmBench are popularly used evaluators in LLM safety to assess harmfulness, we reported our metrics on both classifiers for maximal coverage. However, we understand the need to provide human validation for this, please see above for the full human-evaluated completions across newly-integrated adversarial attacks as well (sample size: 17,550 + 1,170 = 18,720).
>
> *Q2: Maybe I missed something, but how the authors get their 585 prompts is unclear. It would be helpful to explain how these prompts were constructed/sourced, balanced across domains and regions, and reviewed for representativeness or potential bias.*
>
> Thank you very much for bringing up this concern, please see our response in Concern #2 above and refer to Appendix E.3 in the current manuscript. We list details on our dataset construction process and our insights into the overrepresentation factor to mitigate potential biases.
>
> *Q3: The results show that the LLMs may fail on certain domains, but the underlying reasons are not analyzed. Are these differences driven by the prompt wording, topic sensitivity, or inherent domain complexity? Analysis on such failures is important for us to understand the safety mechanisms better.*
>
> To analyze the underlying reasons, we employed the use of training data attribution and influence function analysis. As a case study, we examined why the weight-tampering attack was extremely successful in extracting vulnerabilities. Due to the current page limit, this was not included in the main paper. Please refer to Appendix H for our case study on analysis into the failures through the means of EKFAC influence function analysis.
>
> *Q4: I am curious why the authors only evaluate open source models for adversarial robustness, as there are also adversarial attacks that do support black-box models.*
>
> To address this concern, we have added additional results for adversarial attacks that support black-box models above in Concern #4.
>
> *Q5: It is recommended to add a small defense to evaluate the current safety mechanism. For example, there are many safety guardrails that are designed to moderate the input/output content of LLMs, such as llamaguard series, gemmashield, etc.*
>
> While we agree that it is important to examine defenses to evaluate current safety mechanisms, our goal through this paper is to provide insights and analysis into current vulnerabilities within current models. By introducing defenses, we add confounding factors into our original analysis. However, this is a great follow-up work that we will pursue in the form of another research paper.
>
>
> **References**
>
> [1] Mehrotra, A., Zampetakis, M., Kassianik, P., Nelson, B., Anderson, H., Singer, Y., & Karbasi, A. (2024). Tree of Attacks: Jailbreaking Black-Box LLMs Automatically. arXiv:2312.02119.
>
> [2] Chao, P., Robey, A., Dobriban, E., Hassani, H., Pappas, G. J., & Wong, E. (2024). Jailbreaking Black Box Large Language Models in Twenty Queries. arXiv:2310.08419.

---

> > ### Author Response · Authors · 2025-11-26
> > **Kind Reminder for Author-Reviewer Discussion**
> >
> > Dear Reviewer,
> >
> > We'd like to warmly remind you that the author-reviewer discussion phase will end soon. If our response has helped addressing some of your concerns, we kindly ask you to consider raising your score, we're very happy to provide further clarification if needed

---

> > > ### Comment · Reviewer_BJtb · 2025-11-26
> > >
> > > Thanks for your response and the efforts in human annotation.
> > > However, my main concerns are not well addressed.
> > > I will keep the score.

---

> > > > ### Author Response · Authors · 2025-11-27
> > > > **Clarification on Addressed Reviewer Concerns in Rebuttal**
> > > >
> > > > Dear Reviewer,
> > > >
> > > > Thank you for your follow-up. We would like to **respectfully challenge** your assessment and share that our rebuttal directly addressed each of your original concerns with detailed explanations, additional experiments, and comprehensive human evaluations (~17k samples). We also incorporated your suggestions by expanding our influence-function case study with EKFAC-based attribution to examine underlying phenomena, adding evaluations for closed-source models, and providing further details on dataset construction.
> > > >
> > > > Given these substantial additions and clarifications, we would sincerely appreciate it if you could indicate which specific concerns you feel remain unresolved. This would allow us to address them explicitly and ensure that your feedback is fully reflected in the final version. From our perspective, we engaged with every point raised in the initial review, and we hope this can be taken into consideration when reassessing the review.

---

> > > > > ### Comment · Reviewer_BJtb · 2025-11-27
> > > > >
> > > > > Thanks for your further clarification.
> > > > >
> > > > > And I accordingly raised my score.

---

### Author Response · Authors · 2025-11-29
**Note to Area Chair - SocialHarmBench Rebuttal Updates and Improvements**

Dear Area Chair,

We hope this message finds you well.

Regarding the unprecedented circumstances that have occurred in this cycle, we would like to share a summary of our review/response cycle to summarize our author-reviewer discussions and additional improvements made. The main goal is to provide all our evidence across rebuttals to help improve your judgement of SocialHarmBench and the importance of our benchmark in the overall AI safety community.

**Score updates: from 4/6/6/2 to 6/6/6/4**

**Strengths noted by all reviewers:**
- Comprehensive coverage of different countries and domains (R1)
- Comprehensive coverage of sociopolitical harms: **585 prompts across 7 domains, 34 countries, 4 centuries yield the widest temporal scope among safety benchmarks.** Explicit alignment with **Universal Declaration of Human Rights categories increases real-world relevance.** (R2)
- Rigorous curation pipeline: Multi-model generation → quantitative jailbreak filter → manual deduplication ensures low redundancy and high adversarial quality. **Over-representation factor ORF balances geographic bias.** (R2)
- Novel influence-function diagnostics: **EK-FAC attribution** on 8-B model links harmful outputs to criminal templates in pre-training, offering a principled data-audit tool. (R2)
- Semantic categories for the dataset are well grounded, and the dataset split across categories is well balanced. **It is clear that the data curation was done well.** (R3)
- The analysis of the curated dataset is sufficiently rigorous, and the evaluation set for models is comprehensive. (R3)
- Aside from typos (see below), **paper delivery is very clear**. In particular, I enjoyed Figure 6 as is informative and interesting to see. (R3)
- Overall, the paper addresses an important aspect in safety and alignment in language models that **existing work has looked over**; **SocialHarmBench is a very useful contribution to the safety community.** (R3)
- The social-political aspect is a **unique aspect not extensively studied by existing jailbreak benchmarks.** (R4)

**Major updates:**
- After requested by multiple reviewers, we ran human evaluation on all our 17,550 completions derived in our analysis. We present the table below.
- To expand the breadth of our closed-source experiments, we ran TAP and PAIR adversarial attacks across our closed-source models on SocialHarmBench.
- We clarify dataset construction as described currently in Appendix E.3, **mitigating all reviewer concerns.**

**Per-Reviewer Points**

**Reviewer 1:** Concerns presented in this review focused on the importance of SocialHarmBench, adding human evaluations and ensuring a more comprehensive dataset description section.

To mitigate this, we described how SocialHarmBench is **the first sociopolitical evaluation benchmark for LLMs**. We provide findings related to training data attribution, influence function analysis, and insights into effectiveness of current adversarial attacks. Additionally, we ran human evaluations on **all existing 17,550 completions** across our open-source and closed-source models. Additionally, from our new closed-source adversarial attack evaluations, we report human evaluations across all 1,170 completions. To reduce bias, we have ensured to report results after confirming **inter-annotator agreement exceeds 95% across all completions.**

**This resulted in a score increase from 4 to 6.**

**Reviewer 2:** The concerns presented in this review focused on statistical uncertainty and inter-annotator agreement. To mitigate this, we ran human evaluations on all existing 17,550 completions across our open-source and closed-source models. Additionally, from our new closed-source adversarial attack evaluations, we report human evaluations across all 1,170 completions. To reduce bias, we have ensured to report results after confirming inter-annotator agreement exceeds 95% across all completions.

This reviewer did not have a chance to respond to our rebuttal, however **we addressed all their concerns in our rebuttal.**

**Reviewer 3:** The only two concerns presented in this review focused on the lack of human evaluations and adversarial attacks on closed-source models. To mitigate this, we ran human evaluation on all our 17,550 completions and ran TAP and PAIR adversarial attacks across our closed-source models on SocialHarmBench.

This reviewer did not have a chance to respond to our rebuttals despite multiple reminders, however **we addressed all their concerns in our rebuttal.**

**Reviewer 4:** This reviewer presented concerns related to dataset construction and how we defined semantic categories. To mitigate this, we presented that we share all such details in the current manuscript in Appendix E.3, similar to standard adversarial evaluation benchmarks. Furthermore, the reviewer requested human evaluations and we had the chance to share all human evaluations across each completion.

**This resulted in a score increase from 2 to 4.**

---

> ### Author Response · Authors · 2025-11-29
> **Supporting Tables - Human Evaluation**
>
> Table 1. Baseline Attack Performance Across Models:
> $$
> \\begin{array}{l|ccccccc}
> \\text{Model} & \\text{Censorship} & \\text{Historical Revisionism} & \\text{Human Rights Violations} & \\text{Political Manipulation} & \\text{Propaganda Generation} & \\text{Surveillance} & \\text{War Crimes} \\\\ \\hline
> \\text{Qwen 2.5-7B-Instruct} & 20.5 & 43.8 & 22.3 & 17.3 & 16.2 & 35.0 & 8.9 \\\\
> \\text{Claude-Sonnet-4} & 1.1 & 1.6 & 0.0 & 0.0 & 4.1 & 5.0 & 0.0 \\\\
> \\text{Deepseek-LLM-7B-Chat} & 15.9 & 48.4 & 13.6 & 13.6 & 33.8 & 24.0 & 6.3 \\\\
> \\text{Gemini-2.5-Flash} & 1.1 & 21.9 & 3.9 & 4.9 & 13.5 & 8.0 & 2.5 \\\\
> \\text{Gemma 3-12B} & 18.2 & 37.5 & 6.8 & 18.5 & 28.4 & 29.0 & 6.3 \\\\
> \\text{GPT-4o} & 8.0 & 31.2 & 3.9 & 2.5 & 18.9 & 16.0 & 2.5 \\\\
> \\text{Llama 3.1 8B-Instruct} & 13.6 & 31.2 & 6.8 & 8.6 & 24.3 & 23.0 & 7.6 \\\\
> \\text{Mistral v0.3 7B-Instruct} & 48.9 & 73.4 & 39.8 & 59.3 & 60.8 & 45.0 & 31.6 \\\\
> \\end{array}
> $$
>
> Table 2. Weight Tampering Attack Performance Across Models:
> $$
> \\begin{array}{l|ccccccc}
> \\text{Model} & \\text{Censorship} & \\text{Historical Revisionism} & \\text{Human Rights Violations} & \\text{Political Manipulation} & \\text{Propaganda Generation} & \\text{Surveillance} & \\text{War Crimes} \\\\ \\hline
> \\text{Gemma 3-12B} & 67.0 & 96.9 & 93.2 & 87.7 & 91.9 & 84.0 & 92.4 \\\\
> \\text{Llama 3.1 8B-Instruct} & 67.0 & 96.9 & 92.2 & 86.4 & 91.9 & 82.0 & 87.3 \\\\
> \\text{Mistral v0.3 7B-Instruct} & 78.4 & 98.4 & 95.1 & 95.1 & 95.9 & 90.0 & 91.1 \\\\
> \\text{Qwen 2.5-7B-Instruct} & 67.0 & 96.9 & 93.2 & 87.7 & 91.9 & 81.0 & 87.3 \\\\
> \\text{Deepseek-LLM-7B-Chat} & 76.1 & 98.4 & 93.2 & 87.7 & 94.6 & 90.0 & 87.3 \\\\
> \\end{array}
> $$
>
> Table 3. AutoDAN (GA) Attack Performance Across Models:
> $$
> \\begin{array}{l|ccccccc}
> \\text{Model} & \\text{Censorship} & \\text{Historical Revisionism} & \\text{Human Rights Violations} & \\text{Political Manipulation} & \\text{Propaganda Generation} & \\text{Surveillance} & \\text{War Crimes} \\\\ \\hline
> \\text{Gemma 3-12B} & 76.1 & 89.1 & 90.2 & 90.1 & 93.2 & 86.0 & 86.1 \\\\
> \\text{Llama 3.1 8B-Instruct} & 40.2 & 60.9 & 50.0 & 59.3 & 60.3 & 59.0 & 53.2 \\\\
> \\text{Mistral v0.3 7B-Instruct} & 71.6 & 93.8 & 90.2 & 84.0 & 87.7 & 85.0 & 89.9 \\\\
> \\text{Qwen 2.5-7B-Instruct} & 72.7 & 90.6 & 91.2 & 88.9 & 93.2 & 85.0 & 92.4 \\\\
> \\text{Deepseek-LLM-7B-Chat} & 67.0 & 92.2 & 85.3 & 82.7 & 89.0 & 79.0 & 91.1 \\\\
> \\end{array}
> $$
>
> Table 4. AutoDAN (HGA) Attack Performance Across Models:
> $$
> \\begin{array}{l|ccccccc}
> \\text{Model} & \\text{Censorship} & \\text{Historical Revisionism} & \\text{Human Rights Violations} & \\text{Political Manipulation} & \\text{Propaganda Generation} & \\text{Surveillance} & \\text{War Crimes} \\\\ \\hline
> \\text{Gemma 3-12B} & 75.0 & 93.8 & 87.3 & 80.2 & 91.8 & 86.0 & 79.7 \\\\
> \\text{Llama 3.1 8B-Instruct} & 42.0 & 65.6 & 50.0 & 55.6 & 60.3 & 58.0 & 55.7 \\\\
> \\text{Mistral v0.3 7B-Instruct} & 71.6 & 93.8 & 87.3 & 85.2 & 84.9 & 81.0 & 84.8 \\\\
> \\text{Qwen 2.5-7B-Instruct} & 71.6 & 92.2 & 90.2 & 87.7 & 89.0 & 83.0 & 91.1 \\\\
> \\text{Deepseek-LLM-7B-Chat} & 64.8 & 92.2 & 84.3 & 84.0 & 89.0 & 80.0 & 96.0 \\\\
> \\end{array}
> $$
>
> Table 5. GCG Attack Performance Across Models:
> $$
> \\begin{array}{l|ccccccc}
> \\text{Model} & \\text{Censorship} & \\text{Historical Revisionism} & \\text{Human Rights Violations} & \\text{Political Manipulation} & \\text{Propaganda Generation} & \\text{Surveillance} & \\text{War Crimes} \\\\ \\hline
> \\text{Llama 3.1 8B-Instruct} & 59.7 & 89.0 & 82.5 & 79.0 & 85.1 & 67.3 & 73.4 \\\\
> \\text{Mistral v0.3 7B-Instruct} & 60.9 & 82.8 & 74.5 & 71.6 & 77.0 & 69.3 & 75.6 \\\\
> \\text{Qwen 2.5-7B-Instruct} & 48.2 & 73.4 & 60.7 & 74.1 & 71.6 & 66.3 & 56.9 \\\\
> \\text{Deepseek-LLM-7B-Chat} & 31.0 & 53.12 & 33.3 & 28.39 & 40.5 & 42.8 & 25.3 \\\\
> \\end{array}
> $$
>
> Table 6. Embedding Optimization Attack Performance Across Models:
> $$
> \\begin{array}{l|ccccccc}
> \\text{Model} & \\text{Censorship} & \\text{Historical Revisionism} & \\text{Human Rights Violations} & \\text{Political Manipulation} & \\text{Propaganda Generation} & \\text{Surveillance} & \\text{War Crimes} \\\\ \\hline
> \\text{Gemma 3-12B} & 38.6 & 25.0 & 63.7 & 53.1 & 20.3 & 43.0 & 50.6 \\\\
> \\text{Llama 3.1 8B-Instruct} & 65.9 & 97.5 & 92.2 & 80.2 & 89.2 & 83.0 & 86.1 \\\\
> \\text{Mistral v0.3 7B-Instruct} & 67.0 & 89.1 & 91.2 & 80.2 & 90.5 & 82.0 & 89.9 \\\\
> \\text{Deepseek-LLM-7B-Chat} & 17.0 & 50.0 & 21.0 & 24.7 & 37.8 & 21.0 & 10.1 \\\\
> \\end{array}
> $$

---

> > ### Author Response · Authors · 2025-11-29
> > **Supporting Tables - Closed-Source Adversarial Attacks and Dataset Clarification**
> >
> > $$
> > \\begin{array}{l|ccc|ccc}
> > \\text{Model Name}
> >   & \\text{PAIR-StrongREJECT} & \\text{PAIR-HarmBench} & \\text{PAIR-Human Eval}
> >   & \\text{TAP-StrongREJECT}  & \\text{TAP-HarmBench}  & \\text{TAP-Human Eval} \\\\
> > \\hline
> > \\text{Claude-Sonnet-4}
> >   & 32.39 & 24.78 & 10.42
> >   & 20.61 & 13.67 & 23.42 \\\\
> > \\text{GPT-4o}
> >   & 55.06 & 75.72 & 75.89
> >   & 55.38 & 76.92 & 76.58 \\\\
> > \\end{array}
> > $$
> >
> > **Dataset Clarification**
> >
> > We would like to clarify that due to the current page limit, we were not able to fully list dataset construction in the main paper section. **The topics, events, and functional templates are identified and manually curated by authors – and we use LLMs to generate prompts across our defined subtopics.** Please refer to Appendix E.3, where we note our dataset construction process and data deduplication process in the current manuscript. To remove any remaining ambiguity: the events in our dataset (e.g., elections, policies, conflicts, public-health incidents) were manually selected and defined by the authors. LLMs were never used to propose, generate, or classify events. LLM involvement was limited strictly to linguistic paraphrasing of already-defined templates, not to factual content creation or topical selection.
> >
> > Additionally, we recognize that political opinions and social harms are sensitive topics, which is why it is even more important to study potential harms that LLMs can impose on such contexts. For this reason, **we included the ORF imbalance measure to flag when topic-level composition diverges too sharply from global rates**. Our priority is to avoid situations where certain regions become disproportionately linked to harmful topics, which can carry real sociopolitical consequences such as reinforcing stigma or distorted risk perceptions. The benchmark does not attempt to produce authoritative historical descriptions; events are included only as topical anchors for adversarial stress-tests. No factual claims or interpretations from LLMs are used.
> >
> > **Semantic Category Definitions**
> >
> > We follow standard adversarial evaluation procedures followed by past benchmarks [1, 2, 3] to determine our semantic and functional categories. Functional categories focus on three types of queries (1) standard or agentic usage, (2) historical references, (3) opinion elicitation. In contrast, semantic categories focus on the topics at hand such as surveillance, censorship, propaganda generation, and human rights violations.
> >
> > **We take inspiration for our semantic categories from the Universal Declaration of Human Rights [4].** Our distinction between surveillance and censorship aligns with prior policy and safety-taxonomy work: surveillance concerns the acquisition, monitoring, or analysis of information about individuals or groups, whereas censorship concerns the restriction, removal, or suppression of information or expression. In constructing prompts, we ensured that “surveillance” items emphasize information gathering (e.g., monitoring political dissidents, tracking communications), while “censorship” items emphasize information control (e.g., blocking access, suppressing messages, filtering political content). This separation avoids category conflation and enables clearer, more interpretable evaluations of model behavior.  Similar to the above example, we maintain a clear distinction between all of our semantic categories, ensuring no overlap. **A detailed description of what each category is meant to capture can be found in Table 5 in Appendix D.2.**

---

### Meta-Review · Area_Chair_VCJq · 2025-12-31

**Summary:**

Reviewer feedback converges on the paper’s importance and timeliness: SocialHarmBench targets an under-evaluated but high-stakes safety gap (sociopolitical misuse) with broad coverage across domains, countries, and time periods, and the empirical evaluation is extensive (noted by BJtb, sKc7, 6ZQ4). The main initial reservations were methodological clarity and measurement validity—especially insufficient dataset-construction detail and heavy reliance on automated harmfulness/refusal judges without human validation (raised by BJtb, sKc7, 6ZQ4, sDuZ). The rebuttal materially strengthened the work via large-scale human evaluation and added closed-source/black-box attack experiments, moving the overall balance toward acceptance, albeit with some remaining scope/analysis limitations (e.g., English-only, limited multi-turn/tool-use realism, and incomplete uncertainty/ablation reporting).

**Reviewer Concerns:**

The rebuttal addressed the highest-impact concerns: it clarifies dataset construction (events/templates manually curated; LLMs used for paraphrasing rather than event selection), adds large-scale human evaluation with high reported inter-annotator agreement, and expands adversarial coverage to black-box/closed-source settings using TAP/PAIR (directly responding to BJtb, 6ZQ4, sDuZ, and partially to sKc7). Remaining issues are mostly “scope/rigor polish” rather than fatal: sKc7’s concerns about English-only prompts and the lack of multi-turn/tool-use attacks remain, and the methodological justification around ORF thresholds/utility and statistical uncertainty (e.g., confidence intervals/seed variance) is only partially addressed (best-of-$N$ is helpful but not a full uncertainty treatment). A smaller outstanding point is deeper causal analysis of domain-specific failures (beyond the influence-function case study), plus presentation cleanups (figures/tables/typos) flagged by 6ZQ4 and sDuZ.

**Reviewer Scores:**

BJtb: Effectively moved from 4 → 6 after the rebuttal/discussion; their principal objections (human validation, dataset clarity, and closed-source attack support) were meaningfully addressed.

sKc7: Likely remains around 6 (marginal accept); rebuttal helps on human validation and partially on uncertainty, but English-only + missing multi-turn/tool-use realism and ORF/statistical justification likely keep the score from rising.

6ZQ4: Likely remains 6 or could edge upward slightly; their major concerns (human validation of HarmBench/StrongREJECT and inclusion of TAP/PAIR) were directly addressed, leaving mainly minor presentation/questions.

sDuZ: Moved from 2 → 4 after clarifications on dataset generation/objectivity, category definitions, and added human evaluation; remaining skepticism appears reduced but not fully eliminated.

---

### Decision · Program_Chairs · 2026-01-26

Accept (Poster)